computer modelling and simulation/complexity

crowd, simulation, Turing test

**Author for correspondence:**
Martyn Amos
e-mail: martyn.amos@northumbria.ac.uk

# A Turing test for crowds

## Jamie Webster and Martyn Amos

Department of Computer and Information Sciences, Northumbria University, Newcastle upon Tyne NE1 8ST, UK

MA, 0000-0003-1680-5535

The accuracy and believability of crowd simulations underpins computational studies of human collective behaviour, with implications for urban design, policing, security and many other areas. Accuracy concerns the closeness of the fit between a simulation and observed data, and believability concerns the human perception of plausibility. In this paper, we address both issues via a so-called 'Turing test' for crowds, using movies generated from both accurate simulations and observations of real crowds. The fundamental question we ask is 'Can human observers distinguish between real and simulated crowds?' In two studies with student volunteers ($n = 384$ and $n = 156$), we find that non-specialist individuals are able to reliably *distinguish* between real and simulated crowds when they are presented side-by-side, but they are unable to accurately *classify* them. Classification performance improves slightly when crowds are presented individually, but not enough to out-perform random guessing. We find that untrained individuals have an idealized view of human crowd behaviour which is inconsistent with observations of real crowds. Our results suggest a possible framework for establishing a minimal set of collective behaviours that should be integrated into the next generation of crowd simulation models.

## 1. Introduction

The formal study of human crowds dates back to before the French Revolution [1], but understanding collective behaviour is more urgent than ever before, as populations migrate to high-density urban centres, protests become more organized (and perhaps more common) and increasing numbers of individuals pass through large-scale transportation hubs [2]. A number of computational techniques exist to study the dynamics of crowd behaviour, but the most commonly used is *simulation* [3].

Crowd simulations (generally, but not exclusively, using an agent-based approach) are now employed in many different domains, from events planning and management [4], to urban design [5], and incident response and analysis [6,7]. By studying flows of people *en masse*, and their interactions with the environment and with one another, researchers aim to better understand human collective social behaviour, design more effective and enjoyable public spaces, and improve levels of

safety, security and well-being [8]. An important open question concerns the *believability* of crowd simulations in terms of their macroscopic properties. Put simply, do simulated crowds appear 'lifelike' in terms of their overall behaviour? To answer this question, we took a number of observed crowds moving through a space, and constructed simulations of them that accurately matched the statistical properties of the real crowds (in terms of the number of individuals, clustering, exit choice and so on). We then presented the observed and simulated crowds (both side-by-side and individually), and asked participants to identify the real crowds.

The rest of the paper is organized as follows: in §2, we briefly review related work on crowd simulation and collective behaviour; in §3, we present our 'Turing test' for crowd behaviour, and in §4 we give the results of experimental trials. We conclude with a brief discussion of the implications of our findings.

## 2. Background

The study of human crowd dynamics [9] is motivated by the desire to understand and predict the behaviour of individuals *en masse*, and encompasses a diverse range of crowd types, from large, mainly static crowds at sporting events or concerts [10], to transitory and flowing crowds, such as those found in train stations at rush hour [11], or at religious events such as Hajj [12]. As urban centres grow in size (the United Nations predicts that, by 2050, 68% of the global population will live in cities [13]), we will need to understand and mitigate the impact of crowds on infrastructure, safety, security and quality of life [14]. A number of computational techniques exist to study the dynamics of crowd behaviour, but the most commonly used is *simulation* [3].

Early attempts to understand crowd behaviour were rooted in the physical sciences, using metaphors and mathematical tools from *fluid dynamics* [15], and modelled crowds at the macroscopic level (i.e. without considering individuals) [16]. Subsequent work used an *entity-based* approach, which treated crowds as individual 'particles' [17,18], along with the *agent-based* methodology, in which individuals are treated as semi-autonomous actors [19]. As crowd simulations have become used more frequently, attention has become focused on issues of *accuracy* and *believability*. Here, we define the accuracy of a simulation in terms of its *validity* [20–23]; how closely does the output of the model match data obtained in the real world? It is straightforward to obtain statistical properties of simulation outputs and compare these to the properties of real-world crowds, and that is the approach we take in this paper.

The issue of *believability* is subtly different, and concerns the human perception of whether or not a crowd's behaviour is *plausible*. We emphasize that we do not concern ourselves with 'cinematic' believability (that is, whether or not the rendering of a crowd scene is photo-realistic). Rather, we are interested in whether or not observers are capable of detecting characteristic *patterns of behaviour* that are specific to real crowds, and which may not be present in simulated crowds.

Our testing framework may be thought of as a limited form of the famous 'Turing test', named after Alan Turing, and described in his landmark paper on artificial intelligence [24]. Turing proposed that if a human observer was unable to distinguish between another person and a machine designed to produce human-like responses in a conversational setting, then the machine would be deemed to have 'passed' the test. This type of test has been proposed for biological modelling [25] and artificial life [26] as a way of capturing and interrogating lifelike properties of artificial systems, and assessing the completeness and validity of a model. We base our approach on a related Turing test for collective motion in fish [27].

Our overall aim is to explore how a Turing-like test may be used to examine assumptions and preconceptions about the behaviour of human crowds, and to establish the features of real crowds that must be emulated by a simulation in order for it to be valid and/or 'pass' the test. This is motivated by a widely acknowledged need for crowd simulations to include more 'lifelike' features derived from individual and social psychology (such as group-level behaviours and indecision) [28–30], which are generally not included in software packages, and which give rise to rather unrealistic or 'robotic' patterns of behaviour at the population level. Our experiments represent a first step towards this, by using the Turing test framework to establish distinguishing features of real crowd behaviour.

## 3. Methods

Our experimental methodology was based on that of [27], but with *in-person* (as opposed to online) participants. The first trial tested the ability of participants to distinguish between real and simulated

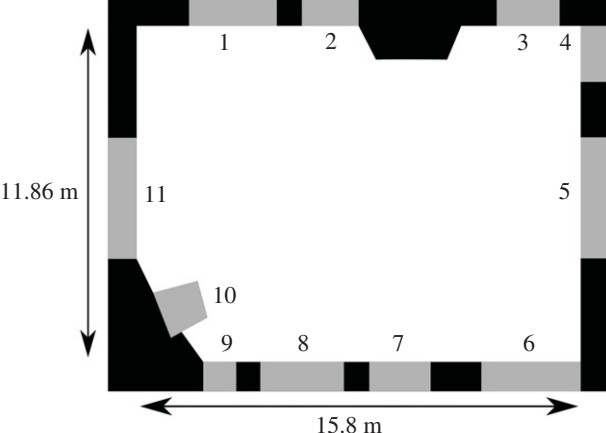

**Figure 1.** Diagram of Edinburgh Informatics Forum (ingress and egress points numbered).

crowds, and the second trial tested the ability of a different set of participants to identify real crowds in a sequence of movies.

For the first trial ($n = 384$), we showed all participants six pairs of randomly ordered movies of 30 s duration (for each pair, both movies were shown simultaneously, side-by-side). In each pair, one movie showed the movement of a real crowd, and the other showed the output of a crowd simulation with the same statistical properties as the real crowd (see §3.5 for more information). Both real and simulated crowd movements were displayed using the same rendering method, and participants were asked to specify on a form which of each pair they thought represented the movement of the *real* crowd. For the second trial ($n = 156$), we used the same set of movies as in the first trial, but this time they were individually presented, in a random order, and participants were asked to classify each movie as either 'Real' or 'Simulated'. In order to prevent possible bias, both sets of participants were disjoint.

## 3.1. Real pedestrian motion dataset

We used data on real pedestrians from the University of Edinburgh School of Informatics [31]. This public dataset, captured in 2010, contains over 299 000 individual trajectories corresponding to the movement of individuals through the School Forum, and is one of the largest open datasets of its type. A diagram of the Forum space is shown in figure 1. The Forum is rectangular in shape (measuring approx. $15.8 \times 11.86$ m), has eleven ingress/egress points, and is generally clear of obstructions. Images were captured (9 per second) by a camera suspended 23 m above the Forum floor, from which individual trajectories were extracted and made available (extraction was performed by Majecka [31]). We note that only the *trajectories* have been made publicly available, and not the original video recordings, for ethical and practical reasons (the image files require several terabytes of storage). However, this dataset has been used in a large number of studies of pedestrian movement/tracking, including [32–34], importantly, none of the individuals whose trajectories were captured were actively participating in movement studies; the trajectories, therefore, are as close to 'natural' as possible (i.e. they have 'behavioural ecological validity' [33]).

In what follows, we use the term 'clip' to specifically refer to a time-limited sequence of trajectory data (whether taken from the Edinburgh dataset or from the output of a simulation), as opposed to a movie visualization. We wrote a utility to search the Edinburgh dataset and extract clips of a specific duration containing a specific number of individuals. This allowed us to ensure that the 'real' and 'simulated' crowds contained the same number of individuals for any single comparison.

## 3.2. Simulation calibration

In order to calibrate our simulation (and, later, to perform statistical analysis), we selected 20 clips at random from the Edinburgh dataset (each of 60 s duration), and calculated the average walking speed of pedestrians observed traversing the Forum. The distribution of speeds is shown in figure 2, with a mean value of $1.17$ m s$^{-1}$. When we simulated these scenarios (see next section), the mean speed of

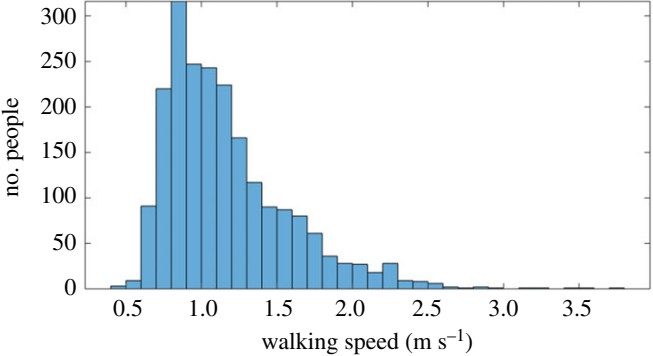

**Figure 2.** Distribution of walking speeds for pedestrians observed in Edinburgh Informatics Forum.

agents was higher ($1.63\,\mathrm{m\,s^{-1}}$), due to the fact that simulated agents were rarely impeded, did not encounter bottlenecks, and were free to accelerate up to their maximum speed. However, as we will see from our results, this did not affect the perception of the simulated crowds.

For the comparison experiments, we randomly selected six clips taken from the Edinburgh dataset (the number of clips is the same as in [27]); each was of 60 s duration, and the number of individuals in a clip ranged from 104 to 194 (with an average of 139). For each clip, we extracted the entry/exit point distribution and the entry time distribution for all individuals. Because we know the locations and dimensions of every 'doorway' in the forum, we were able to easily calculate the entry and exit points for each trajectory in a clip, based on the first and last detected locations. This allowed us to initialize our simulations with the same distributions, ensuring that the runs closely matched the macroscopic properties of the real-world observations (while leaving room for the microscopic differences in which we are interested). We also calculated the average velocity of individuals in each clip, and used this to scale the clip's length (by modifying the video playback speed) to account for variability in camera capture rate, thus normalizing the velocity of individuals relative to expected walking speed [35].

## 3.3. Simulation construction

In order to produce the simulations to accompany each Edinburgh clip, we simulated pedestrian movement using the Vadere package [36]. This package is open-source, which means that (unlike commercial software) its movement models are open to inspection, and it also allows for easy exporting of simulating pedestrian trajectories (which is important when we consider that we must use the same rendering engine for both real and simulated videos).

A crucial component of the simulation is the *crowd motion model*. This defines the rules of interaction between individuals (e.g. avoidance), and between individuals and their environment (e.g. repulsion from walls and physical obstacles), as well as route choice behaviour and differential walking speed. Many different crowd motion models exist [37], but perhaps the most commonly used type is based on *social forces*. Inspired by the fluid flow paradigm of Henderson [15] and others, Helbing and Molnar's social force model (SFM) [38] is a microscopic, continuous model which uses 'attractive' and 'repulsive' force fields between individuals (and between individuals and their environment) to guide movement. The SFM provides the base movement model for a number of pedestrian simulation packages, including FDS + Evac [39], PedSim [40], SimWalk [41] and MassMotion [42], and it has been used extensively in movement research. Additionally, the SFM has been validated using real-world data [20,23], and the comprehensive review of [37] recommends its use in pedestrian movement studies. For all simulations, we used the pre-supplied Vadere template for Helbing and Molnar's SFM, with default attributes and parameters (listed in table 1).

We added small amounts of noise to the simulated trajectories in order to replicate noise in the real crowd data. As the Edinburgh individuals were detected by an overhead camera running at 9 fps, occasional faulty detections caused very short-term errors in the extracted trajectories. Once rendered, this caused individuals to appear to rapidly 'flick' between two headings. As we had no reliable way to quantify the (by inspection, small) amount of noise in the trajectories, we adjusted this by eye until the apparent noise in the simulated data matched the noise level observed in the real data. At any time-step, a simulated agent had a 15% chance of temporarily 'flicking' their heading by a randomly

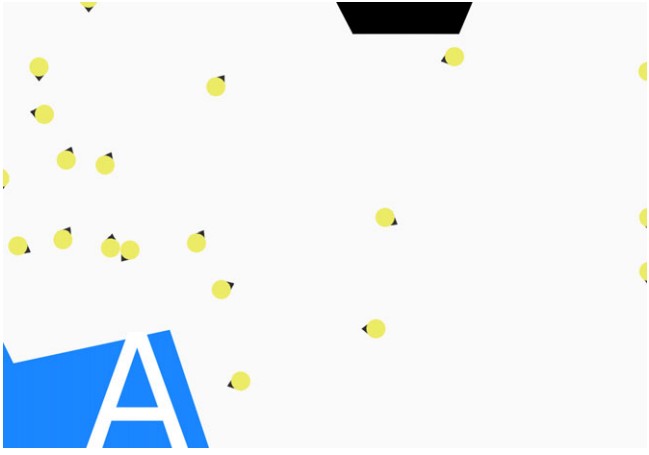

**Figure 3.** Single frame render of an example crowd.

**Table 1.** Vadere simulation model parameters.

| parameter | value |
| --- | --- |
| ODE solver | Dormand–Prince method |
| pedestrian body potential | 2.72 |
| pedestrian recognition distance | 0.3 |
| obstacle body potential | 20.1 |
| obstacle repulsion strength | 0.25 |
| pedestrian radius (m) | 0.2 |
| pedestrian speed distribution mean (m s$^{-1}$) | 1.4 |
| pedestrian minimum speed (m s$^{-1}$) | 0.4 |
| pedestrian maximum speed (m s$^{-1}$) | 3.2 |
| pedestrian acceleration (m s$^{-2}$) | 2.0 |
| pedestrian search radius (m) | 2.0 |

selected value up to 45° (without changing their trajectory). Importantly, as we will see from the results, the addition of this noise had no effect on how the simulated crowds were perceived.

A second artefact of inaccurate detections was that some trajectories had missing sections for several time steps; once rendered, these individuals would temporarily disappear from the frame and then reappear. To fix this, we automatically detected such situations and interpolated coordinates for the missing time-steps when parsing the Edinburgh dataset. We also increased the number of frames per second of both sets of trajectories (real and simulated), from 9 to 72, by interpolating coordinates. This enabled smooth video playback for the purpose of comparisons. Finally, pedestrian trajectories sometimes include 'swaying' motions resulting from the gait of individuals; however, we did not see this in the Edinburgh data.

## 3.4. Crowd rendering

The trajectories of both the simulated and real individuals in each pair of clips were rendered in a uniform fashion, using a tool coded in Java. This allowed us to produce 'top-down' visualizations of both real and simulated clips that were identical in appearance, with individuals represented as filled circles, and headings depicted by an arrow (figure 3).

The use of abstract, simplified shapes and a top-down, two-dimensional presentation is relatively common in crowd studies [10,43–47], although three-dimensional representations are also used [48–51]. We decided against using 'realistic' body shape rendering and three-dimensional views, as initial tests suggested that such a presentation scheme (using animated avatars) would actually

distract viewers from the main aim of the experiment, which was to look for *patterns of behaviour* in the crowd. Additionally, at least one study has shown that crowds that are viewed from the top down are perceived as being just as realistic as those viewed from eye level [52].

## 3.5. Simulation validation

In order to assess the accuracy of our simulations, we calculated several statistical properties for both their outputs and the Edinburgh observations on which they were based. We used two metrics (as in [27]); *polarization* and *nearest-neighbour distance* (NND). The first metric is particularly useful for describing the existence of large groups who might be moving together along the same heading (e.g. from a lecture towards an exit), while the second metric is used for estimating overall crowd density.

Polarization measures the level of 'order' in a crowd, in terms of the heading alignment of members. Polarization is zero when the crowd is completely disordered (everyone is pointing in a different direction), and has a maximum value of 1 when all members of the crowd have the same heading

$$\varphi = \left\langle \frac{1}{N} \left| \sum_N^{i=1} \exp(\iota\theta_i) \right| \right\rangle, \tag{3.1}$$

where $N$ is the number of individuals in the frame, $\iota$ is the imaginary unit, and $\theta_i$ is the heading of each individual.

Nearest-neighbour distance (NND) measures the level of 'clustering' in a crowd. The average NND for a single 'frame' (derived from either the real dataset or the simulation) is calculated from the sum of nearest-neighbour distances of all $N$ individuals

$$\nu = \frac{1}{N} \sum_N^{i=1} d_i, \tag{3.2}$$

where $d_i$ is the nearest neighbour distance between point $i$ and the closest individual in the frame, as calculated by the standard distance formula

$$d_i = \sqrt{(x_2 - x_1)^2 + (y_2 - y_1)^2}. \tag{3.3}$$

In order to confirm that we did not introduce implementation-specific bias by choosing a specific software platform, we compared the outputs of Vadere and JuPedSim [53], an alternative open-source simulation package. We used each package to simulate the 20 real crowd clips mentioned in the previous section, and calculated average NND and polarization over 20 runs for each. The same statistics were then calculated for the real clips (figure 4).

These results confirmed that crowd simulations in Vadere and JuPedSim display similar properties in terms of both NND and polarization, so we selected Vadere as a representative example of crowd simulation packages in general (other reasons for selecting Vadere included the fact that it is written solely in Java, and JuPedSim is built from a combination of C++ and Python, and we found the exporting of agent trajectory data to be more straightforward in Vadere).

Importantly, the statistical properties of the simulations also matched the general properties of the *real* crowds, which confirmed that they are essentially indistinguishable in those terms. In figure 4, we notice a slight difference between the real and simulated crowds in terms of polarization; the real crowds are generally slightly more closely aligned than the simulated crowds, but this difference is of the order of 2%, and we do not believe that this is significant enough to introduce any perceptible difference. Interestingly, the simulation package (as used here) does not use explicit groups of individuals, so the difference we see may be due to the occasional occurrence of polarized groups in the observed data (for example, when a number of individuals all leave a lecture theatre at the same time).

## 3.6. Trial protocol

We performed two trials; the first trial tested the ability of participants to *identify* the real crowds when they were presented alongside the simulated crowds, and the second trial tested the ability of participants to *classify* crowds as either 'Real' or 'Simulated'. We define 'score' in terms of correct identification/classification of the real crowd; so a score of zero means that a participant failed to identify/correctly classify any of the real crowds. All trials took place at the beginning of a class, for which prior permission was obtained from the tutor. Students were informed about the nature of the experiment,

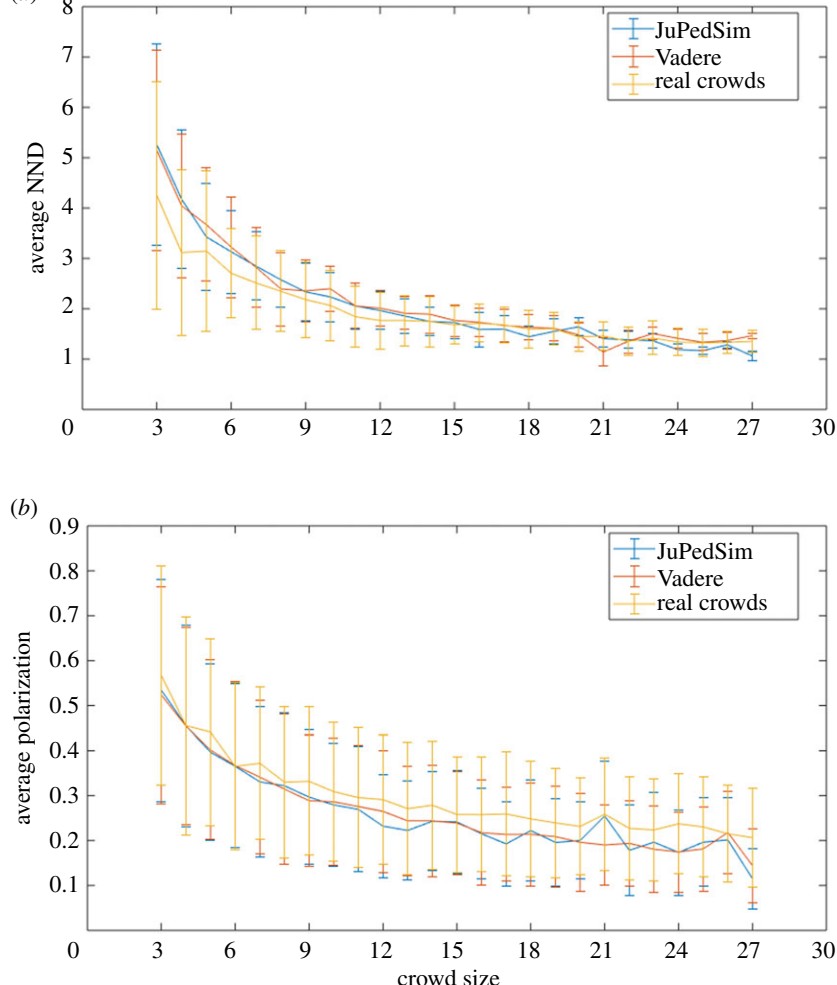

**Figure 4.** Crowd simulations/real crowd statistical comparisons: NND (*a*) and polarization (*b*) as a function of crowd size. The outputs of both simulations have statistical properties that are close to those of the real crowds.

and told that they were under no obligation to participate. Answer sheets were distributed, which consisted of a simple numbered list of tick-boxes. Participants were asked to optionally specify their age and gender. At the end of the trial, participants were also asked to provide some optional narrative notes on any distinguishing features they noticed that allowed them to tell the real crowd from the simulated crowd. Each trial (from initial set-up to collection of answer sheets) took around 10 min.

## 4. Results

### 4.1. Identification trial

We recruited 384 undergraduate students from Northumbria University, distributed over nine groups taken from a mixture of computer science and engineering courses. Of the participants who supplied their details, the gender distribution was 78.83% male, 18.66% female, 2.5% non-binary/other and the average age was 20.7 (we exclude one outlier age value of 71, corresponding to a student's reader).

For each pair, the real and simulated videos were randomly assigned to position A (left) or B (right), and these were combined side-by-side into a single video. Individual videos did not 'loop', and were made up of the first 30 s of the real and simulated crowd clips in each pair. The total duration of the video, showing a total of six comparisons, was 3 min 18 s (including a 3 s pause between each pair). The video is available at http://drives.media/google857, and the real crowds are A, A, B, A, B and B (individual videos are also made available in the electronic supplementary material).

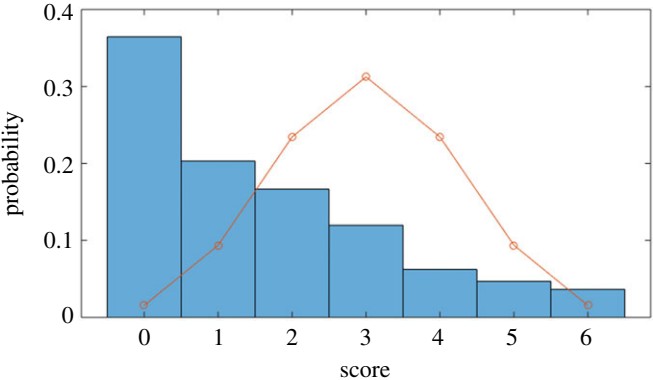

**Figure 5.** Identification trial: distribution of participant scores (the line represents the expected binomial distribution if individuals chose at random).

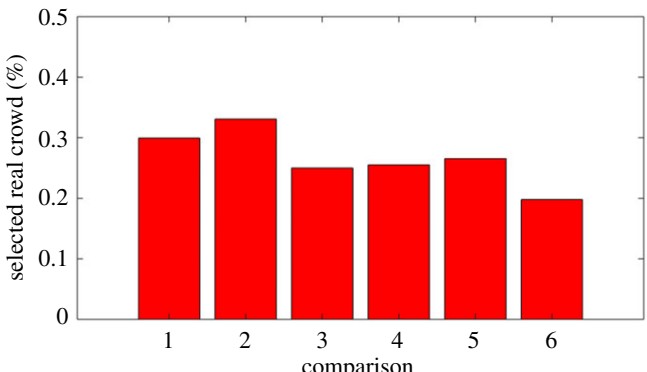

**Figure 6.** Identification trial: success rate across individual pairwise comparisons.

The mean score for participants, across all comparisons, was 1.6 out of 6 (26.7%). That is, participants performed significantly less well than if they had guessed at random. The overall distribution of scores is shown in figure 5, overlaid with the expected binomial distribution (as each comparison is a binary choice, we show this to illustrate the expected distribution of scores if selections were made at random).

If the real and simulated crowds were genuinely indistinguishable (that is, the best strategy would be no better than random guessing), then we would expect roughly 3% of participants (around 12 people) to either guess none correctly, or to guess all six correctly. What we actually found was that over 40% of participants (154 individuals) obtained a score of either zero or six. That is, those individuals were able to correctly *partition* all six pairs of videos into two sets. This answers, in the affirmative, the question concerning the ability of individuals to *distinguish* between real and simulated crowds, even when they have very similar statistical properties.

However, a highly striking result is that the most common score, by far, was zero. A significant proportion of participants (36.46%) failed to identify a single real crowd. Only 3.65% of participants obtained a perfect score of 6. The important implication of this is that participants were reliably able to *partition* videos along the lines of 'real'/'simulated', but were unable to correctly *classify* (that is, label) the crowds. This is a much stronger version of the result obtained in [27], where participants were able to tell real fish from simulated fish, but were not necessarily able to identify the real fish.

We now briefly explore secondary features of our findings for the first trial. The results for each comparison are shown in figure 6, which we present in terms of the proportion of participants who correctly selected the real crowd. These results show that pair 6 presented the strongest challenge to participants, and pair 2 was considered the least challenging. Overall, no clear trend emerged in terms of differential challenge across comparisons.

In terms of variation across groups (figure 7), Group 1 (engineering mathematics students) obtained the most correct identifications, with an average score of 2.46. Group 9 (computer science students) had the fewest correct identifications, with an average score of 1.08 (the remaining groups were all computer science students).

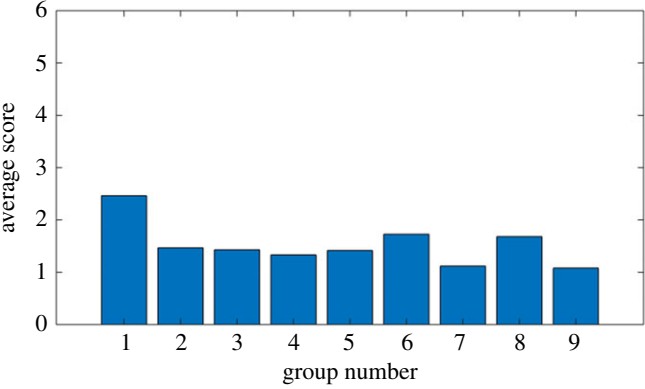

**Figure 7.** Identification trial: distribution of scores across participant groups.

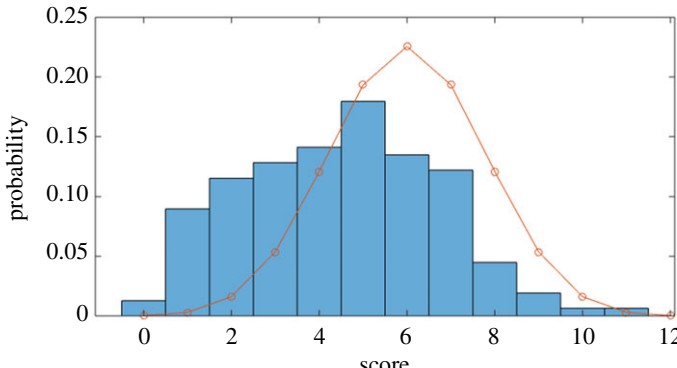

**Figure 8.** Classification trial: distribution of participant scores (the line represents the expected binomial distribution if participants chose at random).

## 4.2. Classification trial

In order to check for bias that might be introduced by showing movies side-by-side, we performed a second trial (with different participants) using exactly the same movies as before; however, this time, the movies were presented *individually*, in a randomized order, and participants were asked to *classify* each one as either 'Real' or 'Simulated'. For this additional trial, we recruited 156 engineering Masters students from Northumbria University, taken from a single unit cohort. Of the participants who supplied their details, the gender distribution was 93.59% male, 3.85% female, 2.56% non-binary/ other and the average age was 23.77.

For this trial, the mean score was 4.47 out of 12 (37.25%; again, much worse than random guessing). The distribution of scores is shown in figure 8, and the success rates for individual movies are shown in figure 9 (the classification order was R(eal), R, R, S(imulated), S, S, S, R, S, R, R, S).

To summarize, we found that the identification task was considerably more difficult when videos were presented side-by-side, as opposed to individually. However, neither mode of presentation allowed participants to perform better than random guessing. Taken together, the results of both trials suggest that off-the-shelf simulations 'fail' the crowd Turing test, in that they actively 'mislead' untrained experts into believing that real crowds are actually simulated, and vice versa. We now consider why this might be.

## 4.3. Narrative findings

In this section, we analyse the free text supplied by participants. We focus, in particular, on the large number of participants in the first trial who scored zero, as (a) they consistently misidentified the real crowd, and (b) the number of comments supplied by participants in the classification trial was relatively low, and they were generally consistent with the comments made by participants in the first trial. We highlight themes and specific comments that may shed light on the assumptions and

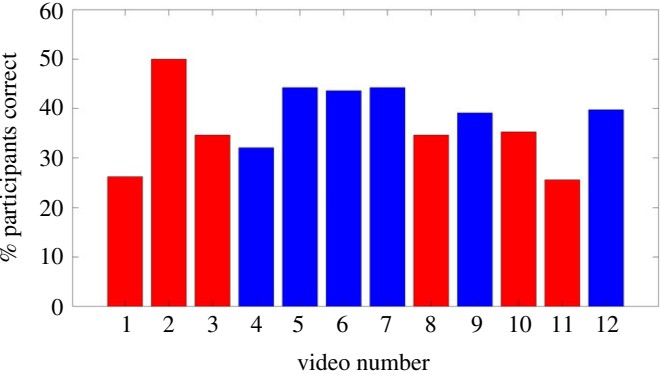

**Figure 9.** Classification trial: success rate for each movie, expressed as the percentage of the population who correctly classified it as real or simulated. Real crowds are shown as red bars, simulated crowds as blue bars.

preconceptions held by these individuals, that led them to consistently 'flip' the real and simulated crowds in their perception.

The first theme that emerged concerned rapid or 'random' *changes of movement* in the real crowd, which many participants incorrectly attributed to the simulated (fake) crowd. Versions of this included 'Fake changed direction too quickly', 'Fast change suggests fake', 'Generated crowd had too much random movement', 'Real seemed to change direction gently'. Although the average speed of the simulated agents was higher than that of the real people, participants singled out rapid movement in the *Edinburgh* (i.e. real) videos as indicative of artificiality (when, in fact, the real people moved more slowly). Overall, 72 participants mentioned a variant of this type of observation. The underlying assumption here is that real people move smoothly, at a uniform speed, and do not tend to deviate much from their chosen path.

A second common theme concerned *avoidance*; many participants incorrectly assumed that real people would avoid close contact with one another, whereas the simulated individuals would 'overlap' or collide. Representative quotes included 'Simulated people collided, real crowds avoided each other', and 'People overlapping'. In reality, the opposite is true, as the real dataset contains multiple instances of individuals coming into close proximity. Moreover, the social forces model *explicitly* tries to keep individuals *apart* unless close proximity is unavoidable, so the behaviour (distance keeping) that participants attributed to real people was actually an in-built feature of the simulation. This theme was mentioned by 22 participants.

Perhaps the most profound observations concerned *perceived intentionality* and *group-level behaviour*; many participants believed that 'On the whole, people have relatively smooth and intentional paths' (this was actually a feature of the simulation), 'Real crowds don't really stand around' (stationary groups were only present in the real dataset), and 'The real ones knew where they were going' (this was actually a function of the simulation's path choice algorithm). Variations on this theme were mentioned by seven participants. The interesting thing here is that participants (incorrectly) ascribed clear human intentionality and purpose to the *simulated* agents (Real crowds move more purposefully), and failed to acknowledge it in the actual humans that were observed.

Overall, we found that participants believed that individuals in crowds are orderly, purposeful, respectful of personal space, and consistent and uniform in their speed and direction. In fact, all of these characteristics were features of the simulation. Participants also failed to recognize features of real crowds, such as rapid changes in speed or direction, close proximity of individuals, and stationary groups/individuals, all of which were discounted by participants as being 'glitchy' or 'unrealistic'.

## 5. Conclusion

In this paper, we presented a Turing test for crowds that allowed us to investigate issues of believability in crowd simulations by comparing them with visualizations representing data obtained from real pedestrians. We performed trials with over 500 university students, and found that, while the students were generally able to discriminate between 'real' and 'artificial' crowds, they were unable to correctly label them.

We acknowledge several potential limitations of our study; the use of students as test subjects is the subject of ongoing debate [54], and the computer science background of many of the students (and the gender imbalance) may have biased our results. It may be the case that our students have become conditioned to make certain assumptions about how crowds behave from playing games that use a relatively unrealistic crowd model. However, this is merely speculation on our part. Nonetheless, an important future development of this work will be to re-run the trials using experts in crowd dynamics, to find out whether they are better placed to identify the real crowds (based on our findings, we would recommend a sequential rather than side-by-side presentation mode). This is entirely consistent with Harel's expectation of how a biological modelling Turing test might work; '… our interrogators can't simply be any humans of average intelligence. Both they and the … people responsible for "running" the real organism and providing its responses to probes, would have to be experts on the subject matter of the model, appropriately knowledgeable about its coverage and levels of detail' [25].

If (as we might expect) the experts are able to reliably identify the real crowd, then this immediately suggests a mechanism for ascertaining the minimal set of crowd features that are necessary to 'pass' the test. If, for example we identified that 'group-level movement' was a 'flag' for the experts, we might include such a behaviour in the simulation and re-run the trial with a second group of experts. If the experts are then less able to tell the difference between real and simulated crowds, then we might conclude that group-level behaviour constitutes an important feature that should be included in simulations. This would represent a formalized methodology for implementing a number of recommendations that have been recently made by a number of crowd scientists, who call for the integration into software of a wider range of psychological and interpersonal processes [28–30]. These recommendations reflect a pressing need to revisit physics-based models of crowd behaviour which, though they may generate macroscopic behaviour that is reasonably realistic, fail to capture the inherent 'messiness' and unpredictability of real human crowds.

Ethics. Informed consent was obtained from all participants, and the study protocol was approved by the Northumbria University Faculty of Engineering and Environment Ethics Board, application no. 16433.

Data accessibility. The datasets, code and movies supporting this article have been uploaded as part of the supplementary material available at https://doi.org/10.6084/m9.figshare.c.4859118.v1.

Authors' contributions. M.A. and J.W. conceived and designed the study, performed the trials, analysed the results and drafted the manuscript; J.W. wrote the software and carried out the computational analysis.

Competing interests. The authors declare no competing interests.

Funding. This work was supported by partial PhD studentships sequentially awarded by the Manchester Metropolitan University Faculty of Science and Engineering and the Northumbria University Faculty of Engineering and Environment.

Acknowledgements. We thank Jeremy Ellman for advice on analysis, and colleagues at Northumbria University for assistance with participant recruitment.

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
