## [Reviewer comments · Royal Society Open Science]

Review History

RSOS-200307.R0 (Original submission)

Review form: Reviewer 1

Is the manuscript scientifically sound in its present form?

Yes

Are the interpretations and conclusions justified by the results?

Yes

Is the language acceptable?

Yes

Do you have any ethical concerns with this paper?

No

Have you any concerns about statistical analyses in this paper?

No

Recommendation?

Accept with minor revision (please list in comments)

Comments to the Author(s)

Dear authors and editor,

On the whole, I think this is a really interesting piece of work. It does have limitations, but these are acknowledged and openly discussed.

Some of the future work the authors mention in the conclusions would help greatly to better understand the results, but even as it stands, this manuscript presents a useful contribution to research in this field.

I only have a few minor comments on aspects of the manuscript that would benefit from an enhanced discussion in my view. Otherwise I am happy for this work to be published.

Kind regards.

****Comments****

p. 4, line 8: please provide details on how the route choice distribution was extracted.

When obtained by filming from overhead, pedestrian trajectories typically include 'swaying' that results from individuals shifting their weight from one foot to the other. Did you not find this in your data? If you did find it, discuss the effect it could have on how people view the believability of different clips.

Please explain why you used polarisation and NNDs to compare the general properties of the real crowd and the simulations. For ref. [21] this makes sense, as fish shoals tend to be cohesive and NND has biological relevance (adjusted in response to predators to reduce risk), but I am not convinced this is the case for pedestrian crowds.

NND may be relevant for estimating densities and the distribution of people in the environment. But polarisation is surely largely driven by the direction of routes and route choice.

Other measures may be more relevant, such as ones relating to congestion, paths walked to avoid others... .

I think a more detailed discussion on this (perhaps in the discussion or an appendix would be useful).

In the caption of fig. 5, specify that you show the binomial distribution one would expect to find if people decide randomly (as you have done in the text).

I'm a bit confused about your discussion of the findings in figure 5. A 0 score suggests that someone has failed to identify any of the real crowds correctly. That means they consistently believe the simulated crowd to be the real crowd. As such, they can classify, but they have their labels wrong. This is different to your statement 'but most of them were unable to say which was which', because if that was the case, we would have expected peaks in scores for both 0 and 6. I agree that your result is stronger than the findings in [21].

The first theme you discuss as a possible explanation for your findings hints at an issue with the data (the flips in movement direction). It's unfortunate this could not be removed from the empirical data.

Review form: Reviewer 2

Is the manuscript scientifically sound in its present form?

Yes

Are the interpretations and conclusions justified by the results?

Yes

Is the language acceptable?

Yes

Do you have any ethical concerns with this paper?

No

Have you any concerns about statistical analyses in this paper?

No

Recommendation?

Accept with minor revision (please list in comments)

Comments to the Author(s)

The authors address the challenge of recognizing real collective human behavior dynamics through a Turing test adapted to recognize between real motion of agents and simulated motion of agents. The agents represent real pedestrians from the University of Edinburgh School of Informatics. The Turing test has two phases: Identification and Classification. In Identification, simulated and real behavior is presented in two parallel videos. In Classification, the same videos are presented randomly in sequence. The participants, student volunteers, must then identify whose videos represent real or simulated crowd behavior. The results show that, in Identification, the students were able to differentiate the set of videos, but not able to say precisely which one was the real or the simulated. In Classification the results were a little bit better, but still worse than random choice.

The study is very interesting and the results and narratives collected are very important to gain awareness about behavior idealization that can bias and create inaccurate models of collective human behavior.

I think the paper has a clear message and it is well written. I just have small observations.

- While describing the demographics of the participants, I think it would complement well the manuscript adding the information about the specific courses of each group.
- For the Identification videos, the authors say which are the ones with the real behavior, but for the Classification that is not mentioned. Since the videos are put available in a given sequence, that information would fit in the manuscript too.
- Even though it seems not to affect the results, I wonder if a closer mean speed to the real data would highlight other types of narrative obtained by the participants, since there are some videos that when put side by side show very different velocity motions.
- I wonder if having side by side simulations with the same amount of agents (that doesn't seem to be the case) would yield other results.
- If Vadere and JuPedSim have similar statistics, it was not clear if there was a specific reason for choosing Vadere.
- "Our findings would, therefore, appear to contradict [29], in which the authors state that "However, people do more than just walk. (...)" Since the simulated crowds are able to mimic that and it is the narrative findings that contradicts that, I think this could be rephrased for a more clear sense of the sentence.
- The authors also discuss the academic background of the participants. A last question/suggestion is about gender bias. Do you think that with a more balanced sample there would be possible to evaluate if there is a gender more prone to get the high scores?

Decision letter (RSOS-200307.R0)

Dear Professor Amos

On behalf of the Editors, I am pleased to inform you that your Manuscript RSOS-200307 entitled "A Turing Test for Crowds" has been accepted for publication in Royal Society Open Science subject to minor revision in accordance with the referee suggestions. Please find the referees' comments at the end of this email.

The reviewers and handling editors have recommended publication, but also suggest some minor revisions to your manuscript. Therefore, I invite you to respond to the comments and revise your manuscript.

- Ethics statement

- Data accessibility

If you wish to submit your supporting data or code to Dryad (<http://datadryad.org/>), or modify your current submission to dryad, please use the following link:
<http://datadryad.org/submit?journalID=RSOS&manu=RSOS-200307>

- Competing interests

- Authors' contributions

AB carried out the molecular lab work, participated in data analysis, carried out sequence alignments, participated in the design of the study and drafted the manuscript; CD carried out the statistical analyses; EF collected field data; GH conceived of the study, designed the study,

coordinated the study and helped draft the manuscript. All authors gave final approval for publication.

- Acknowledgements

- Funding statement

Because the schedule for publication is very tight, it is a condition of publication that you submit the revised version of your manuscript before 19-Jun-2020. Please note that the revision deadline will expire at 00.00am on this date. If you do not think you will be able to meet this date please let me know immediately.

If your manuscript is newly submitted and subsequently accepted for publication, you will be asked to pay the article processing charge, unless you request a waiver and this is approved by Royal Society Publishing. You can find out more about the charges at <https://royalsocietypublishing.org/rsos/charges>. Should you have any queries, please contact openscience@royalsociety.org.

on behalf of Professor Jon Crowcroft (Associate Editor) and Marta Kwiatkowska (Subject Editor)
openscience@royalsociety.org

Associate Editor Comments to Author (Professor Jon Crowcroft):

Comments to the Author:

there are a few suggestions from the reviewers for clarification which I believe would improve the paper, though I don't think extra work is needed, just some wordsmithing....

Reviewer comments to Author:
Reviewer: 1

Comments to the Author(s)

Dear authors and editor,

On the whole, I think this is a really interesting piece of work. It does have limitations, but these are acknowledged and openly discussed.

Some of the future work the authors mention in the conclusions would help greatly to better understand the results, but even as it stands, this manuscript presents a useful contribution to research in this field.

I only have a few minor comments on aspects of the manuscript that would benefit from an enhanced discussion in my view. Otherwise I am happy for this work to be published.

Kind regards.

****Comments****

p. 4, line 8: please provide details on how the route choice distribution was extracted.

When obtained by filming from overhead, pedestrian trajectories typically include 'swaying' that results from individuals shifting their weight from one foot to the other. Did you not find this in your data? If you did find it, discuss the effect it could have on how people view the believability of different clips.

Please explain why you used polarisation and NNDs to compare the general properties of the real crowd and the simulations. For ref. [21] this makes sense, as fish shoals tend to be cohesive and NND has biological relevance (adjusted in response to predators to reduce risk), but I am not convinced this is the case for pedestrian crowds.

NND may be relevant for estimating densities and the distribution of people in the environment. But polarisation is surely largely driven by the direction of routes and route choice.

Other measures may be more relevant, such as ones relating to congestion, paths walked to avoid others...

I think a more detailed discussion on this (perhaps in the discussion or an appendix would be useful).

In the caption of fig. 5, specify that you show the binomial distribution one would expect to find if people decide randomly (as you have done in the text).

I'm a bit confused about your discussion of the findings in figure 5. A 0 score suggests that someone has failed to identify any of the real crowds correctly. That means they consistently believe the simulated crowd to be the real crowd. As such, they can classify, but they have their labels wrong. This is different to your statement 'but most of them were unable to say which was which', because if that was the case, we would have expected peaks in scores for both 0 and 6. I agree that your result is stronger than the findings in [21].

The first theme you discuss as a possible explanation for your findings hints at an issue with the data (the flips in movement direction). It's unfortunate this could not be removed from the empirical data.

Reviewer: 2

Comments to the Author(s)

The authors address the challenge of recognizing real collective human behavior dynamics through a Turing test adapted to recognize between real motion of agents and simulated motion of agents. The agents are represent real pedestrians from the University of Edinburgh School of Informatics. The Turing test has two phases: Identification and Classification. In Identification, simulated and real behavior is presented in two parallel videos. In Classification, the same videos are presented randomly in sequence. The participants, student volunteers, must then identify whose videos represent real or simulated crowds behavior. The results show that, in Identification, the students were able to differentiate the set of videos, but not able to say precisely which one was the real or the simulated. In Classification the results were a little bit better, but still worse than random choice.

The study is very interesting and the results and narratives collected are very important to gain awareness about behavior idealization that can bias and create inaccurate models of collective human behavior.

I think the paper has a clear message and it is well written. I just have small observations.

- While describing the demographics of the participants, I think it would complement well the manuscript adding the information about the specific courses of each group.
- For the Identification videos, the authors say which are the ones with the real behavior, but for the Classification that is not mentioned. Since the videos are put available in a given sequence, that information would fit in the manuscript too.
- Even though it seems not to affect the results, I wonder if a closer mean speed to the real data would highlight other types of narrative obtained by the participants, since there are some videos that when put side by side show very different velocity motions.
- I wonder if having side by side simulations with the same amount of agents (that doesn't seem to be the case) would yield other results.
- If Vadere and JuPedSim have similar statistics, it was not clear if there was a specific reason for choosing Vadere.
- "Our findings would, therefore, appear to contradict [29], in which the authors state that "However, people do more than just walk. (...)" Since the simulated crowds are able to mimic that and it is the narrative findings that contradicts that, I think this could be rephrased for a more clear sense of the sentence.
- The authors also discuss the academic background of the participants. A last question/suggestion is about gender bias. Do you think that with a more balanced sample there would be possible to evaluate if there is a gender more prone to get the high scores?

Author's Response to Decision Letter for (RSOS-200307.R0)

See Appendix A.

Decision letter (RSOS-200307.R1)

Dear Professor Amos,

It is a pleasure to accept your manuscript entitled "A Turing Test for Crowds" in its current form for publication in Royal Society Open Science.

on behalf of Professor Jon Crowcroft (Associate Editor) and Marta Kwiatkowska (Subject Editor)
openscience@royalsociety.org

Associate Editor Comments to Author (Professor Jon Crowcroft):

Associate Editor

Comments to the Author:

Thanks for the revised submission and for addressing the reviewers comments.

Appendix A

We thank the reviewers for their extremely positive and helpful comments. We now describe how we have addressed these in the revised manuscript.

Reviewer 1

p. 4, line 8: please provide details on how the route choice distribution was extracted.

Added a description of how these were extracted to Section 3(b), and clarified that it was *entry/exit point* distributions that we calculated (as the simulator dynamically calculates the *route* based on these).

When obtained by filming from overhead, pedestrian trajectories typically include 'swaying' that results from individuals shifting their weight from one foot to the other. Did you not find this in your data? If you did find it, discuss the effect it could have on how people view the believability of different clips.

We did not see any noticeable "swaying" in the raw trajectory data; this may be due to the relatively high positioning of the camera above the Forum; trajectories were also smoothed by the interpolation step performed prior to rendering, so we do not believe that this has affected believability.

Please explain why you used polarisation and NNDs to compare the general properties of the real crowd and the simulations. For ref. [21] this makes sense, as fish shoals tend to be cohesive and NND has biological relevance (adjusted in response to predators to reduce risk), but I am not convinced this is the case for pedestrian crowds. NND may be relevant for estimating densities and the distribution of people in the environment. But polarisation is surely largely driven by the direction of routes and route choice. Other measures may be more relevant, such as ones relating to congestion, paths walked to avoid others... . I think a more detailed discussion on this (perhaps in the discussion or an appendix would be useful).

We included polarisation in order to capture groups who might be moving together along the same heading (e.g., from a lecture towards the exit). Interestingly, the simulation package provides no explicit functionality for such groups, and we saw slightly higher polarization values in the "real" data compared to the simulated data, which could be explained by the occasional occurrence of such groups in the observed data. As the reviewer observes, NND is useful for capturing density. We have added some additional text to Section 3(e) to clarify this.

In the caption of fig. 5, specify that you show the binomial distribution one would expect to find if people decide randomly (as you have done in the text).

Done.

I'm a bit confused about your discussion of the findings in figure 5. A 0 score suggests that someone has failed to identify any of the real crowds correctly. That means they

consistently believe the simulated crowd to be the real crowd. As such, they can classify, but they have their labels wrong. This is different to your statement 'but most of them were unable to say which was which', because if that was the case, we would have expected peaks in scores for both 0 and 6. I agree that your result is stronger than the findings in [21].

Changed to "The important implication of this is that participants were reliably able to *partition* videos into "real"/"simulated", but were unable to correctly *classify* (that is, label) the crowds."

The first theme you discuss as a possible explanation for your findings hints at an issue with the data (the flips in movement direction). It's unfortunate this could not be removed from the empirical data.

This is, indeed, unfortunate; as we state in the paper, we have tried to compensate for this noise in the data by artificially adding noise to the simulated trajectories, so that both "real" and "simulated" agents perform these rapid short-term "flicks" with roughly the same frequency. However, we do not believe that this had any significant impact on the results.

Reviewer 2

- While describing the demographics of the participants, I think it would complement well the manuscript adding the information about the specific courses of each group.

Added clarification to Section 4(a) that all other groups were CS.

- For the Identification videos, the authors say which are the ones with the real behavior, but for the Classification that is not mentioned. Since the videos are put available in a given sequence, that information would fit in the manuscript too.

We do state this in the caption to Figure 9 (real crowds are red, simulated crowds are blue), but we have added additional clarification text in the main body.

- Even though it seems not to affect the results, I wonder if a closer mean speed to the real data would highlight other types of narrative obtained by the participants, since there are some videos that when put side by side show very different velocity motions.

Producing simulations with a closer mean speed to the real data would require that we make modifications to agent behaviour, as the default behaviour of Vadere (and other simulators) is for agents to only deviate slightly from the average walking speed parameter. In the real crowds people are seen running, standing still, or with a mix of velocities in a single trajectory. These behaviours are not present by default in most simulation packages, and it is precisely this sort of difference between simulated and real behaviour that we are trying to investigate (basically, over-fitting the behaviour of the simulator to the observed data would mean that we were not performing a fair comparison with the default simulator, as one of the main points of this project is to establish precisely what modifications are needed to make a simulator more realistic...)

- I wonder if having side by side simulations with the same amount of agents (that doesn't seem to be the case) would yield other results.

Simulations were calibrated to include roughly the same number of people as the real crowd in each pair; however, the simulations *appear* to contain fewer people in most comparisons, as the only goal of the agents is to navigate the forum and leave the room. People in the real data tended to spend longer in the forum than in the simulations, as people often stop in groups and frequently change their destination.

The amount of time spent in the forum per agent may be another useful metric to investigate in planned follow-on work, when we introduce new behaviours to the simulation model.

- If Vadere and JuPedSim have similar statistics, it was not clear if there was a specific reason for choosing Vadere.

Added clarification to Section 3(e); basically, Vadere is written solely in Java, with which we are comfortable, and its exporting of trajectory data for agents is better (in our opinion).

- "Our findings would, therefore, appear to contradict [29], in which the authors state that "However, people do more than just walk. (...)" Since the simulated crowds are able to mimic that and it is the narrative findings that contradicts that, I think this could be rephrased for a more clear sense of the sentence.

On reflection, we have deleted this sentence, as we now do not believe that it conveys the precise message we were trying to get across (and, as the reviewer states, it slightly confuses the main issue).

- The authors also discuss the academic background of the participants. A last question/suggestion is about gender bias. Do you think that with a more balanced sample there would be possible to evaluate if there is a gender more prone to get the high scores?

We did not see any clear difference in the scores between genders; however over 80% of our participants were male, so we cannot confidently evaluate the ability for any gender to get higher scores unless we were to repeat the trial with a more balanced sample.